# Balancing the Strength–Impact Relationship and Other Key Properties in Polypropylene Copolymer–Natural CaSO_4_ (Anhydrite)-Filled Composites

**DOI:** 10.3390/ijms241612659

**Published:** 2023-08-10

**Authors:** Marius Murariu, Fouad Laoutid, Yoann Paint, Oltea Murariu, Jean-Marie Raquez, Philippe Dubois

**Affiliations:** 1Laboratory of Polymeric and Composite Materials, Materia Nova Materials R&D Center & UMONS Innovation Center, 3 Avenue Copernic, 7000 Mons, Belgium; fouad.laoutid@materianova.be (F.L.); yoann.paint@materianova.be (Y.P.); oltea.murariu@materianova.be (O.M.); 2Laboratory of Polymeric and Composite Materials, Center of Innovation and Research in Materials and Polymers (CIRMAP), University of Mons (UMONS), Place du Parc 20, 7000 Mons, Belgium; jean-marie.raquez@umons.ac.be

**Keywords:** polypropylene (PP), impact polypropylene copolymers, mineral-filled composites, calcium sulfate, gypsum and anhydrite II, thermal and mechanical properties, reactive extrusion, REx, automotive and engineering applications

## Abstract

To develop novel mineral-filled composites and assess their enhanced properties (stiffness, a good balance between mechanical strength and impact resistance, greater temperature stability), a high-impact polypropylene copolymer (PPc) matrix containing an elastomeric discrete phase was melt mixed with natural CaSO_4_ β-anhydrite II (AII) produced from gypsum rocks. First, in a prior investigation, the PPc composites filled with AII (without any modification) displayed enhanced stiffness, which is correlated with the relative content of the filler. The tensile and impact strengths dramatically decreased, especially at high filling (40 wt.%). Therefore, two key methods were considered to tune up their properties: (a) the ionomeric modification of PPc composites by reactive extrusion (REx) with zinc diacrylate (ZA), and (b) the melt mixing of PPc with AII surface modified with ethylenebis(stearamide) (EBS), which is a multifunctional processing/dispersant additive. The properties of composites produced with twin-screw extruders (TSEs) were deeply assessed in terms of morphology, mechanical, and thermal performance, including characterizations under dynamic mechanical solicitations at low and high temperatures. Two categories of products with distinct properties are obtained. The ionomeric modification by Rex (evaluated by FTIR) led to composites characterized by remarkable thermal stability, a higher temperature of crystallization, stronger interfacial interactions, and therefore noticeable mechanical properties (high tensile strength (i.e., 28 MPa), increased stiffness, moderate (3.3 kJ/m^2^) to good (5.0 kJ/m^2^) impact resistance) as well as advanced heat deflection temperature (HDT). On the other hand, the surface modification of AII with EBS facilitated the dispersion and debonding of microparticles, leading to composites revealing improved ductility (strain at break from 50% to 260%) and enhanced impact properties (4.3–5.3 kJ/m^2^), even at high filling. Characterized by notable mechanical and thermal performances, high whiteness, and a good processing ability, these new PPc–AII composites may be tailored to meet the requirements of end-use applications, ranging from packaging to automotive components.

## 1. Introduction

Polypropylene (PP) is one of the most used and cheapest thermoplastics available today, with growing demands for various end-use markets (packaging, textiles, building and construction, automotive, electrical, and engineering components) [1,2,3,4]. Produced as PP homopolymers (PPhs) or copolymers (PPcs), PP offers a combination of outstanding physical, mechanical, thermal, and electrical properties (not encountered in other thermoplastics), excellent processability, and the possibility of recycling [4,5,6]. Moreover, adequately modified, or customized, PP is a low-cost, performant material of interest for many engineering applications, such as in the automotive industry, which is considered one of the largest PP end-use sectors. Characterized by a good strength-to-weight ratio in relation to other materials (i.e., metals), having higher functionality and more economic manufacture, stability at high temperatures, etc., the list of applications in the automotive area includes various interior and exterior components [4,7,8,9]. Still, regarding the current developments of PP, it is important to mention the recent announcement of the production of bio-based PP made from eco-friendly building blocks using natural resources [10].

Nowadays, following the progress in the technology of production of PP, three main categories of PP products are commercialized, which are characterized by the dissimilar composition of the polymer chains: (a) PP homopolymers (PPhs), produced through the polymerization of propylene; (b) PP random copolymers, produced through the polymerization of propylene with low amounts of ethylene, traditionally up to 5–6% (or other comonomers); and (c) impact copolymers, or ‘heterophasic’ block copolymers, which are mainly obtained through the copolymerization of propylene and ethylene (ethylene content is much higher than in random copolymers) [11]. PPhs are characterized by higher crystallinity and improved chemical resistance compared to PPcs; they are stiffer and have better strength at high temperatures. Unfortunately, the use of PPhs as engineering thermoplastic materials is limited because of their poor impact resistance at room temperature and low temperatures [12]. Therefore, several approaches have been considered to improve PP toughness. The melt blending of PP with various impact modifiers (ethylene–propylene rubber (EPR), ethylene–propylene–diene monomer (EPDM), styrene–ethylene–butylene–styrene (SEBS), etc.) was the earliest route for producing impact-modified PP; the copolymerization of propylene with α-olefinic moieties was also applied, whereas other methods are described in the state of the art [13,14]. The copolymerization of propylene with ethylene by means of sequential reactors was proven to be extremely effective and a low-cost method to obtain high-impact-resistant PPc (more information in Section 2.1), which is commonly referred to as impact-PPc [13,15]. The low T_g_ (typically in the range of −40 to −50 °C) of the dispersed ethylene–propylene rubber (EPR) within the propylene-rich phase allows the dramatic improvement of the impact strength at low and room temperatures. Compared to PPhs, PPcs are characterized by increased flexibility and durability while preserving a good chemical resistance and gas barrier properties. They can be easily transformed by different processing methods such as injection molding (IM), extrusion (to produce films, fibers, sheets), blow molding, thermoforming, etc. The applications of PPc products are defined by their properties, and they range from packaging, through pipe and medical items, to components for the automotive industry, as well as for mechanical and electrical parts [14].

The mineral-filled PP composites can offer enhancements over unfilled PP (homopolymers or copolymers) in properties such as stiffness, thermal and dimensional stability, heat deflection at higher temperatures, etc., and in many cases, better processing and cost effectiveness [16]. To expand the property profile of products, PP has been melt-blended with different mineral fillers: clays [17,18], CaCO_3_ [19,20,21,22], talc [23,24,25], mica [26,27], kaolin [25,28,29], barium sulfate [30], etc. Nonetheless, only a limited number of laboratory-scale studies have considered the use of synthetic CaSO_4_ (abbreviated after this CS) [31,32].

It is also worth mentioning that the mineral fillers are typically polar/hydrophilic in nature, while the PP matrix is apolar/hydrophobic. Therefore, for improved properties, several techniques are considered for polymer matrix–filler interfacial compatibilization: the use of fillers with various surface treatments (with surfactants and chemical coupling agents [33]), increasing the relative polarity of the PP matrix and reactivity (i.e., using maleic anhydride-grafted polypropylene (MAgPP) [34,35], by incorporation of ethylene ionomers/polar copolymers [36,37], and other methods [38].

Recently, we have drawn attention to the utilization of so-called “insoluble” anhydrite, which is a filler derived from natural or synthetic gypsum; it is characterized by high thermal stability and advanced whiteness (meaning that it can be used as a TiO_2_ extender). Unfortunately, this filler is not well known in the industry of polymer composites. Compared to other unstable forms of CS, the calcination of gypsum at high temperatures (e.g., at 500–700 °C in an industrial process) allows obtaining highly stable β-anhydrite II (AII) [39,40,41]. It is also important to restate that AII made from synthetic or natural gypsum has been used with promising results and without adverse effects on the polymer matrix, particularly for the manufacturing of polylactic acid (PLA) composites [39,40,42].

Related to this study, which valorizes and develops the results of an applied R&D project, it is worth mentioning that nowadays, the producers of natural gypsum are looking for new markets and new applications (polymer composites, paints, etc.). The use of natural gypsum and its derivatives (AII) can be an attractive option to produce new PP composites and an answer to current industrial requests.

However, it was reported by us in an earlier study [43] that the high filling of a PPh with AII led to PP composites characterized by increased rigidity and thermal stability, but also to a significant decrease in tensile strength and impact resistance (dramatically diminishing), especially at higher filler content. Therefore, “custom” compositions have been produced via reactive extrusion (REx) to obtain PP composites with significant enhancements in properties.

On the other hand, linked to the novelty of this study, it is assumed that the properties of PP composites are largely determined by the nature of the PP matrix, i.e., PPh or PPc. Owing to the higher impact resistance of the PPc, the melt blending of the PPc with rigid fillers appears to be a choice of great interest, considering the large availability of the raw materials. Moreover, in some cases, modified inorganic particles can serve as stress concentrators to build up the stress field around the filler itself, contributing to the consumption of the impact energy; therefore, a better toughness is obtained [44].

Going from current industrial requests, the main goal of this paper is to show the experimental pathways followed to produce and characterize novel PPc composites filled with up to 40 wt.% AII (made from natural gypsum) for use in applications requiring materials assessing improved stiffness, a good balance between mechanical strength and impact resistance (NB: better than using a PPh as the polymer matrix), enhanced thermal properties, and other specific benefits (good processing, whiteness, low cost).

For a prior evaluation, small quantities of PPc–AII composites have been produced first by melt blending using internal mixers, which is an experimental step followed by the characterization of the main properties. Correlated with the amounts of filler (20–40 wt.% AII, filler without any modification), the mechanical rigidity (Young’s modulus) of PPc composites showed some enhancement, whereas an important reduction was evidenced for their tensile and impact strengths, especially at high AII loading, i.e., 40 wt.%.

The reactive modification of PPc–AII compositions with metallic ionomeric additives (i.e., zinc diacrylate—ZA) has been once more considered to obtain more performant polymer composites [43]. On the other hand, a second compatibilization approach has been followed as well. To offer better value in composites, usually the fillers are “coated” and/or surface treated to improve the mechanical properties and moisture resistance, reduce surface energy and melt viscosity, and enhance dispersion and processing characteristics. Traditional surface modifiers such as fatty acids (e.g., stearic acid) and their salts, silanes, titanates, zirconates, etc., are generally indicated for the treatment of mineral fillers [16].

To produce higher quantities of PPc–AII composites and to process them by IM, the PPc was filled by melt compounding with 20–40 wt.% of previously modified AII, using twin-screw extruders (TSEs). For better filler dispersion and compatibility, or stronger interfacial adhesion, two approaches have been considered, respectively, following the physical/surface modification of AII microparticles with ethylenebis(stearamide) (EBS) [45,46] or using reactive metallic ionomeric additives (i.e., ZA) [36,43]; both methods were reported in prior studies to improve the properties of polymeric composites. Then, the effects of filling PPc with up to 40 wt.% AII in these custom composites were deeply evaluated in terms of morphology, mechanical, and thermal properties to evidence their key-properties for potential/further applications. Interestingly, the presence of EBS on the surface of micro-filler in PPc–AII composites was found to limit the drastic reduction in the strain at the break and of Izod impact resistance (at high filling), the values remaining much higher than those of the level reference (i.e., >4 kJ/m^2^). 

On the other hand, following the ionomeric modification (evaluated by FTIR spectroscopy), due to the formation of clusters/ionic crosslinks of Zn ionomer and favorable interactions with the filler, a higher stiffness, better tensile strength, and flexural properties were obtained together with a more spectacular increase in thermal stability and heat deflection temperature (HDT).

These results indicate the key role of the polymer matrix and of interfacial properties, highlighting the benefits of special modifiers like ZA and EBS, which are paving the way for two categories of products with distinct characteristic features, in the specific case of PPc/AII composites. The whiteness and good processability by IM are additional features of these new composite materials; therefore, they can be considered of interest for a large segment of applications, going from conventional utilization (packaging) to engineering products.

## 2. Results and Discussion

For an easier understanding of the results discussed hereinafter, it will be very helpful to look at the information given in the experimental section regarding the techniques considered in this study to produce and characterize the properties of PPc–AII composites (see elsewhere, Section 3). For simplicity, the PP copolymer used as a polymeric matrix will be abbreviated hereinafter as “PPc”.

For primary evaluation, the PPc–AII composites have been produced in small quantities using internal mixers (see experimental part), whereas in a second investigational step, custom compositions (with AIIt or AII-ion) were produced with TSEs.

### 2.1. Prior Characterizations: PPc Matrix and Composites Produced with Internal Mixers

As mentioned in the introductory part, it is important to restate that the dehydration of natural gypsum (CaSO_4_·2H_2_O) and calcination at high temperature allow obtaining so-called “insoluble” β-anhydrite II (AII), which is characterized by excellent thermal stability, high whiteness, and an extremely slow rate of rehydration even after the prolonged immersion time in water [40,41].

For more information about this inorganic filler and to refrain from repetition, we recommend consulting some recent papers published by us regarding the use of AII to produce novel polymer composites [40,43] or relevant references [41].

Regarding the polymer matrix used in this study (PPc), this copolymer is traditionally produced at industrial scale using a sequential system with two consecutive reactors. The isotactic PP (iPP) homopolymer is made in a first reactor and transferred to a second one, where ethylene and propylene comonomers are copolymerized to create ethylene propylene rubber (EPR) in the form of microscopic nodules dispersed through the iPP phase [47]. The in situ production leads to a heterophasic structure (EPR as a discrete phase) inside of a semi-crystalline PP homopolymer matrix. 

However, the composition and morphology of high-impact PPcs are complex, because they are usually assimilated to some kind of blend containing both ethylene–propylene (EP) random copolymers, EP block copolymers with different sequence lengths, and isotactic PP homopolymers (iPP) [48]. Compared to iPP homopolymers, these high-impact PPcs have moderate tensile strength and stiffness (NB: improvements can be obtained by filling). The elastomeric phase characterized by low T_g_ is responsible for the important increase in PPc impact strength with respect to PPh, both at room temperature and at low temperatures [13,15].

#### 2.1.1. Morphology of PPc Matrix

To obtain more insight into the morphology of the polymer matrix used in this study, SEM analyses (in SE mode) have been performed on the impact fractured surfaces of unfilled PPc samples (Figure 1a,b). The presence on the fractured surfaces of microscopic nodules of EPR and of circular micro-voids (mean dimension around 1.1 µm) formed after the removal or debonding of EPR from the iPP matrix is clearly revealed by the SEM micrographs. Interestingly, the EPR nodules are uniformly dispersed through the iPP matrix and, accordingly, will play a key role in increasing the impact strength, as confirmed by mechanical characterizations. Indeed, it was reported that the dispersed rubber particles from the PPc could deform in a ductile manner during mechanical solicitations and/or generate crazes that absorb the impact energy, preventing the growth of catastrophic cracks [49]. It is worth pointing out the additional presence of some platelet-like structures visible in SEM pictures (Figure 1a), which can be reasonably attributed to the presence of specific additives used by the supplier to modify PPc, e.g., nucleating agents (NAs).

#### 2.1.2. Composites Produced with Internal Mixers: Mechanical Properties

The results of mechanical characterizations of PPc–AII composites produced with internal mixers and processed by CM are shown in Table 1. The abbreviation for composite samples is as follows: PPc–xAII, where x is the loading (wt.%) of AII in composites; PPc–xAII-ion, where “ion” denotes ionomeric modification, i.e., with 2% ZA.

It is worth mentioning that in the first experiments, the filler (AII) was used without any modification. The progressive increase in filler percentage to 40 wt.% in composites leads to the gradual decrease in the tensile strength at yield (σ_y_) from 27 MPa (unfilled PPc) to around 20 MPa for the highly filled composites (PPc–40AII). In contrast, the mechanical rigidity (Young’s modulus) of composites increases, which is well correlated with the amounts of AII, e.g., by about 55% for PPc–40AII. However, the elongation at the break (ε_b_) decreases drastically, especially at high filling, from 44% (unfilled PPc) to about 9% for PPc composites containing 40% AII.

The characterization of impact performances, a key parameter followed in this work due to its relevance for automotive applications, highlighted the excellent impact properties of the PPc matrix, i.e., Izod impact resistance/strength of 7.6 kJ/m^2^. Unfortunately, at high loadings (PPc–40AII sample), the impact resistance is reduced more than two times (i.e., to 3.1 kJ/m^2^), denoting an important drop with respect to the level reference considered as an objective for the targeted application, meaning values higher than 4 kJ/m^2^. The decrease in mechanical properties (i.e., tensile strength, impact resistance, etc.) at high loadings is traditionally attributed to the formation of aggregates due to particle/particle interactions, the creation of stress-concentration points, low matrix/filler interactions, ineffective wetting of the filler by the polymer, insufficient adhesion and homogenization, and so on [50].

Therefore, to compensate for the decrease in both tensile and impact strengths after the high filling of PPc with 40 wt.% AII, it was necessary to propose custom formulations, such as the reactive modification with ZA. As illustrated in one key example in Table 1, by comparing PPc–40AII to PPc–40AII-ion samples, it is confirmed that the ionomeric modification is a promising practice that could lead to more performant composites: σ_y_ is boosted, respectively, from 20 to 27 MPa, while the impact resistance is remarkably improved from 3.1 to 4.7 kJ/m^2^.

The above results led to the conclusion that custom compositions are required to enhance the properties of PPc/AII composites, e.g., via reactive modification with ZA ionomeric additives. The formation of zinc salt clusters has been reported to take place within the polyolefin matrix through (i) thermally triggered free-radical grafting of ZA acrylate double bond(s) onto the PPc chains and (ii) the in situ formation of thermoreversible ionic networks as sketched in Figure 1 [51]. In the presence of inorganic (polar) particles such as AII, the Zn–ionomer clusters are thus expected to improve the compatibility and interfacial interactions between the ionomeric PPc and filler, as has been published elsewhere [43], with a concomitant increase in both tensile strength and impact toughness. Hence, due to their potential interest in engineering applications, these custom PPc–AII–ion composites have been proposed for upscaling by REx using TSEs.

On the other hand, a second compatibilization approach has been considered as well. It consists of surface treating AII with 1% EBS (see the experimental part) followed by dispersion within PPcs using TSEs. Indeed, it has been reported elsewhere that the surface treatment of (nano)fillers with EBS offers better filler dispersion and enhanced processing ability, but also significant improvements in the impact strength of the so-produced polymeric (nano)composites [46].

### 2.2. PPc–AII Composites Produced with Twin-Screw Extruders (TSEs)

To produce higher quantities of PPc–AII composites and to process them by injection molding (IM), it has been decided to carry out the melt blending and filler dispersion with a twin-screw extruder (TSE). Table 2 shows the codes and compositions of PPc composites produced with TSEs that are discussed in this section. For more clarity, it is important to add that AIIt denotes AII “treated” with EBS, whereas “AII-ion” was used for the filler/formulations with ZA ionomeric additive, as modified by REx. Hereinafter, we discussed the most important results connected to the key properties of these novel PPc composites, which are filled with either AII-ion or AIIt. For the sake of comparison, unfilled PPc (processed under similar conditions) is used as a reference.

#### 2.2.1. FTIR Analysis of Ionomeric Modified PPc Composites

FTIR spectroscopy is known as a powerful tool of analysis to provide important information about the ionomeric modification of composites following different methods of production, i.e., in this study with internal mixers or TSEs. Figure 2 shows the comparative FTIR spectra of the raw materials (ZA, PPc and AII) and those of PPc–AII-ion composites obtained by REx. At this point, it is important to remember that the PPc matrix is supplied with specific additives (nucleating, anti-static, and other agents) that can lead to a more difficult interpretation of the spectra. Consequently, for a particular wavelength range (1700–800 cm^−1^), here are some general statements connected to the reactive modification of PPc with ZA (able to produce thermoreversible ionic crosslinking, as well as covalent bonds with the polyolefin chains, vide supra), and to the identification of AII particles in composites. Regarding the unfilled PPc, with only one exception (an absorption band around 1260 cm^−1^ ascribed to the presence of a specific additive), all FTIR signals/bands specific to PP are identified in agreement with reported data [52,53]. The evidence of ethylene–propylene sequences, particularly for PPc but not for PPh [54], was confirmed by a specific band at 720 cm^−1^.

ZA exhibited multiple characteristic absorption bands, as assigned in Figure 2. It shows a strong absorption band at 1650 cm^−1^ assigned to stretching vibrations of C=C bonds (in conjugation with C=O groups) [55], at 1593 and 1532 cm^−1^ attributed to the asymmetric stretching of the Zn carboxylate anion, and also at 1437 cm^−1^ assigned to the symmetric stretching of strongly coupled carboxylate anion from ZA [56,57,58]. At lower wavenumbers, ZA shows additional absorption bands, e.g., at 1371 cm^−1^ (attributed to the –CH_2_– wagging [56] but also often described as δas(COO^−^) [59]; there is not a consensus in the literature), at 1277 cm^−1^ (peak assignment to α,β-unsaturated carboxylate), at 968 and 826 cm^−1^ (assignments to out-of-plane C−H bending vibration from =CH_2_) [57]. Analyzing the spectra of PPc–AII-ion composites produced by REx, the absence or the strong attenuation of the signal at 1650 cm^−1^, thus assigned to C=C bonds from the bifunctional ZA, is reasonably ascribed to the free-radical reaction of ZA (via its carbon–carbon double bond) and PPc chains mostly through their methine groups, leading to the formation of covalent bonds, i.e., of Zn ionomer. Moreover, the peaks at 968 and 826 cm^−1^ are not detected anymore after the reactive extrusion of PPc with AII/ZA. However, analyzing the spectra of PPc–AII-ion samples in the region from 1600 to 1500 cm^−1^, only bands of lower intensity are visible at around 1540 cm^−1^, which likely correspond to the carbonyl groups/carboxylate stretching of the zinc carboxylate [57,60]. Considering that the loading of ZA in composites is relatively low (2 wt.%), we will refrain from some more comments from the perspective of more comprehensive forthcoming studies on this topic. On the other hand, the spectrum of AII shows a broad band above 1100 cm^−1^, which is assigned to the presence of (SO_4_)^2−^ groups. This strong band is also detected in PPc–AII-ion composites (at about 1120 cm^−1^), which also show supplementary/distinct peaks at ca. 1150 cm^−1^ that are specific for the calcinated gypsum at high temperature [61,62]. To be complete, it is worth pointing out that the analysis of the FTIR spectra of composites obtained either with internal mixers or REx led to very similar conclusions (Appendix A).

#### 2.2.2. Morphology of Composites

First, it is important to mention that the granulometry of AII microparticles is characterized by a D_v50_ = 3.6 µm and D_v90_ = 12.9 µm (SEM images and more information are in Section 3.1, Materials). For better evidence of the filler distribution throughout the PPc matrix, SEM imaging was performed using the backscattered electron mode (BSE) to obtain a higher phase contrast linked to the composition of the samples. In fact, the BSE technique has a higher sensitivity to differences in atomic number; therefore, it is very useful for giving valuable information about the distribution of AII microparticles within the PPc matrix (i.e., the presence of Ca atoms is evidenced by brighter zones). Figure 3a–h shows representative SEM-BSE micrographs at different magnifications performed on the cryo-fractured surfaces of the PPc composites. Conventionally, it is expected to obtain better individual particle dispersion at a lower filler amount, i.e., at 20% AII (Figure 3b,f) than at a high filling level (40%). Somewhat surprising, quite well-distributed particles, with various geometries and broad size distribution, are evidenced at the surface of cryo-fractured PPc composites, even at high filler loading. Regarding the differences in morphology, only some SEM images suggest a slightly better dispersion using AIIt than AII-ion particles (e.g., Figure 3d vs. Figure 3h).

As it was reported elsewhere [43], SEM combined with energy-dispersive X-ray spectroscopy (SEM-EDX) may also be used as a technique of analysis (results not included here) to obtain supplementary information about the dispersion of fillers (i.e., AIIt versus AII-ion). In fact, SEM/EDX can reveal the elemental atomic distribution within the PPc matrix of some elements of interest, such as Ca, S and O from filler, Zn from the ionomeric modifier, and others. Still, from the SEM micrographs performed at higher magnification, it comes out that the AII particles are characterized by a relatively low aspect ratio and an irregular shape of micrometric size.

#### 2.2.3. Thermal Properties

Traditionally, the fillers improve the thermal stability of polymer composites [63]. Obviously, the thermal treatments of gypsum (CaSO_4_·2H_2_O) at high temperatures (up to 700 °C) lead to a filler (AII) characterized by excellent thermal stability. Therefore, in the absence of undesired catalytic degradation of the polymer matrix, it is expected that the incorporation of AII into different polymers will lead to composites characterized by similar or even better thermal stability. The addition of AII into different polymer matrices (e.g., PLA, PP, others) was reported to lead to a delay in their thermal degradation (a significant stabilization effect), which can be quite well distinguished by the increase in the maximum decomposition temperatures (T_d_) [39,40,43]. Indeed, from the comparison of TG and D-TG traces (Figure 4a,b) of unfilled PPc and PPc composites, it comes out that the filling with AII is responsible for a significant delay in the thermal degradation of PPc matrix, which is well correlated with the amounts of filler (AIIt or AII-ion). Moreover, an impressive stabilization effect is obtained following the reactive modification with ZA for PP–AII-ion composites, which is clearly highlighted by the increases recorded of both thermal parameters T_5%_ and T_d_, by nearby 60 °C and 80 °C, respectively, in comparison to the unfilled PPc (Table 3). Such improvements in thermal properties offer the possibility of processing at a higher temperature, for instance, by injection molding (IM).

Regarding the thermal properties as revealed by DSC analysis (Figure 5a,b, and Table 4), it is important to remind again that the PPc matrix is supplied with special additives, including a nucleating agent able to increase the rate and temperature of crystallization, and to control the size of spherulitic structures. Hereinafter, the DSC results following the non-isothermal crystallization (DSC cooling) and the subsequent heating scans are compared, both at 10 °C/min (Figure 5a and Figure 5b, respectively).

Concerning thermal behavior as evidenced during cooling, a similar temperature of crystallization (T_c_) is noted for the unfilled (nucleated) PPc and PP–AIIt composites (T_c_ of 122–123 °C). There is no evidence of any additional effect on the values of T_c_, which are associated with the presence of AII and/or EBS (T_cEBS_ = 138 °C) from AIIt. However, it is reasonable to attribute these results to the effectiveness of the nucleating agent (NA), which is added in PPc by the supplier, and to the good crystallization ability that is generally specific for PPc. On the other hand, even though PPc contains a NA, the ionomeric modified composites (PP-AII-ion) show a significant increase in T_c_ up to 129 °C, apparently without some influence linked to the relative amount of filler. The increased T_c_/rate of crystallization of PPc is ascribed to the formation of Zn ionomeric crosslinks, and this behavior can be beneficial for the rapid cooling to allow shorter processing cycles in industrial conditions.

To be complete, one can also observe a very small exothermal peak at ca. 102 °C, which is more readily visible for the unfilled PPc and PPc–AIIt composites (as highlighted by black arrows in Figure 5), which is ascribed to the presence of some crystallizable component from the heterophasic PPc, such as the crystallization of PE-rich sequences of low molecular weights [15] (T_m_ at 118 °C in the second DSC heating), or to the crystallization of some EPR components [49]. However, it is not totally excluded that this can be accounted for processing additives as well. From the second DSC heating scans (Figure 5b), only minor changes in T_m_ (165–167 °C) are observed, whereas by comparing the enthalpies of melting and the degree of crystallinity (Table 4), they are somewhat higher in the case of composites with respect to the unfilled (nucleated) PPc.

#### 2.2.4. Mechanical Properties

The mechanical properties of particulate-filled polymer composites are determined to a great extent by the performance and nature of the polymer matrix, the interfacial adhesion between matrix and filler, and the filler shape, size, and relative content [64]. The tensile and impact properties of unfilled PPc and PPc composites are depicted in Figure 6a–c. 

By analyzing the effects of AIIt addition into PPc, it comes out that the ultimate (yield) tensile strength (σ_y_) of the PPc (29 MPa) is only moderately decreased at high filling (40% AIIt), i.e., down to 25 MPa. Interestingly, this value is significantly higher than that obtained using AII without any modification (i.e., σ_y_ of 20 MPa) in experiments with internal mixers and therefore for specimens processed by CM. The better distribution/dispersion of AII microparticles following the modification with EBS (as highlighted by SEM micrographs), and the enhanced compatibility of the filler with the PPc matrix, can explain in part the improvements in the tensile properties of these composites.

On the other hand, from the perspective of their utilization in engineering applications, the most interesting performances in terms of tensile strength are ascribed to PP–AII-ion composites (σ_y_ of 28 MPa), apparently without any clear evidence connected to the influence of AII loadings. In fact, it comes out that for stronger interfacial adhesion between the polar polymer matrix (PPc) and the hydrophilic filler (i.e., AII), the modification of PPc can be accomplished using metallic (Zn) ionomers, which is an experimental approach paving the way to more performant composites from the point of view of stiffness and tensile strength. Nevertheless, for applications requiring ductility, good/moderate tensile strength, and increased impact toughness (Figure 6c), the use of an AII surface treated by EBS can be an option of great interest.

It is also important to mention that the strain at break (ε_b_) of PPc–AII-ion composites drastically decreased (ε_b_ is lower than 13%) when compared to PPc (ε_b_ above 330%) and PPc–AIIt composites (ε_b_ of 260% and close to 60%, at, respectively, 20 and 40 wt.% AIIt). The reduction in ε_b_ in the case of PPc–AII-ion composites is mainly ascribed to the effects of reinforcing with rigid AII microparticles and to the formation of ionic thermoreversible crosslinks (Zn ionomer), which are able to promote stronger interfacial interactions, without totally excluding the contribution of other factors.

Regarding the evolution of rigidity/Young’s modulus (E), the recorded values are primarily determined by the amount of filler (Figure 6b). Compared to the unfilled PPc (E of 1400 MPa), the highly filled composites (PPc–40AIIt and PPc–40AII-ion) are characterized by advanced rigidity (E of 2100 MPa and 2200 MPa, respectively), which finally represents an enhancement of more than 50% with respect to PPc.

For additional insight regarding the interfacial properties and behavior of PPc composites in tensile tests, SEM analyses were performed on the surfaces of specimens fractured during tensile testing (Figure 7a–f). The unfilled PPc is characterized by very high ductility, and therefore, the SEM micrographs are evidencing numerous elongated and laminated regions or filaments/fibrils (Figure 7a,b). Furthermore, the abundant holes/micro-voids are attributed to the presence of the discrete elastomeric phase (EPR), which is inherently less resistant to mechanical stress. Regarding the composites filled with AIIt (Figure 7c,d), it is noted that the well-dispersed microparticles through the PPc matrix (typically of lower dimensions, Figure 7c) allowed the better preservation of matrix ductility even at high filling levels. In addition to the better dispersion of the filler, it is also assumed that following the surface treatment of AII with EBS, the strength of the matrix/filler adhesion is decreased. A similar behavior is reported elsewhere for AII treated with stearic acid [65]. Moreover, in the case of PPc–AIIt composites (evaluated by tensile and impact tests), the debonding is considered the dominant micromechanical deformation process, whereas the local plastic deformation remains much larger around the treated particles (AIIt). On the other hand, stronger surface adhesion at the PPc–AII interface can be ascribed to PPc–AII-ion composites (together with the decrease in ductility). This hypothesis is supported by the SEM images (e.g., Figure 7f) that highlight the presence of numerous zones of intimate/cohesive contact at the PPc–AII interface, which can explain the higher tensile strength of these composites.

Regarding the impact properties of specimens produced by IM, PPc, being reported as an impact (block) copolymer, is characterized by a good Izod impact resistance (as high as 6.7 kJ/m^2^) allowing its utilization in applications requiring high impact toughness, such as automotive components, but also for freezer and freezer-to-microwave applications. Generally, it is stated that the reinforcement of PP with inorganic fillers leads to the improvement of some mechanical properties, especially of the stiffness; i.e., higher tensile/flexural modulus, and enhanced flexural strength are obtained. On the other hand, the reduction in the strain at break and impact strength is often reported [66,67]. Talc, which is the most commonly used inorganic filler for the production of PP composites due to its reinforcing and nucleating effects, unfortunately, at high amounts leads to dramatic decreases in impact strength because of the anisotropic nature of this filler [13].

Interestingly, to formulate new bio-based materials for automotive applications, Notta-Cuvier et al. have characterized, for comparative purposes, a commercially available 20% talc-filled compound based on a PP block copolymer [68]. The composite had an impact resistance (Izod) of 4.8 kJ/m^2^. In the present study, PPc has been filled with similar amounts of filler (i.e., 20% AIIt or AII-ion), the composite materials displaying slightly better impact resistance, i.e., in the range of 5.0 to 5.3 kJ/m^2^ (results depicted in Figure 6c). Moreover, by increasing the amounts of filler at 40%, only the PP-40AIIt composites keep a good impact resistance at the level of 4.3 kJ/m^2^, whereas a reduction is noted in the case of PP-40AII-ion composites (3.3 kJ/m^2^). Interestingly, even at high loadings of filler, the impact strength of PP-40AII-ion composites remains superior to that reported for the neat PPhs (1.6–2.3 kJ/m^2^ [43,49]). They are of interest, especially for applications requiring high tensile strength, stiffness, low deformation and increased dimensional stability.

On the other hand, the failure behavior of the material at a higher strain rate during impact testing differs from those under tensile stress at a lower strain rate. The crazing is the main deformation mechanism considered in the case of impact solicitation rather than the shear yielding [49]. The outstanding impact properties of PPc and PPc composites are firstly ascribed to the presence of the EPR phase (vide supra). The presence in all SEM images (Figure 8a–f) of round-shape micro-voids is due to the debonding and release of EPR nodules from the PP phase. Following the surface modification of AII with EBS, AIIt has been reported to contribute to a better preservation of the impact toughness because of the partial or total debonding of AII microparticles from the matrix (Figure 8d), which additionally consumes some mechanical energy from the impact solicitation. In addition, by comparing the SEM images of PPc–40AIIt and PP–40AII-ion composites (Figure 8d vs. Figure 8f), in the latter case, the filler looks more anchored in the polymeric matrix, i.e., with less visible debonding of AII-ion microparticles. Such an observation can be interpreted once more as evidence of the stronger interfacial properties using AII-ion than AIIt.

A better interfacial adhesion (PPc–filler) using AII-ion, with respect to AIIt, has been further attested by the evaluation of flexural properties (Figure 9). Like in tensile tests, the flexural modulus is increasing correlated with the relative amount of filler, with the most important rise recorded for PP–AII-ion composites (e.g., by more than 120% upon the addition of 40% filler). On the other hand, regarding the maximum/ultimate flexural strength, somewhat surprisingly, the values for PPc–AIIt composites are almost comparable to those of unfilled PPc, and this can be an additional indication for the reduced interfacial adhesion, i.e., for the lower transfer of the mechanical stress at the PPc–AIIt interface. On the other hand, following the ionomeric modification, important increases are obtained for the flexural strength of PPc–AII-ion composites (by more than 30% with respect to PPc). This difference can be again ascribed to the better interfacial adhesion between the filler-polymeric matrix as modified with Zn ionomer.

#### 2.2.5. Dynamic Mechanical Analysis (DMA)

DMA has been used to provide information about the viscoelastic properties of PPc composites under mechanical cyclic deformation over a high range of temperatures, i.e., from −75 to +150 °C, and about the temperatures of transition previously not identified using the DSC method. Figure 10a,b shows the comparative evolution of the (a) dynamic storage modulus (E′) and (b) loss modulus (E″) of unfilled PPc and PPc composites.

E′, which is often associated with the ‘stiffness’/elastic component of the materials [69], increased distinctly in all temperature ranges for all composites and proved dependent upon the relative amount of filler (Figure 10a). This trend is in line with the data recorded for the evolution of the Young’s modulus in tensile tests, actually emphasizing the reinforcing effect of the filler. The substantial increase in E′ is linked to the possibility of using these composites at higher mechanical stress and increased temperatures.

Figure 10b shows the evolution of E″ (purely viscous component) as a function of temperature. E″ displays for all samples three regions of transition indicating the increase of the internal friction and changes in the moving of PPc chain segments [70]: at ca. −45 °C, which is assigned to T_g_ of the EPR component, and at 12–14 °C, which is ascribed to the second-order transition temperature of the isotactic PP homopolymer (iPP) component [71], and it is often attributed to the β-relaxation of unrestricted amorphous iPP (T_g_ of iPP) [72,73]. Still, the third transition seen below 70 °C is ascribed to the α_c_ relaxation of rigid amorphous iPP [72], and it is frequently associated with the polymer chain rearrangements [74]. It is also worth mentioning that the damping factor (Tan δ, E″/E′ ratio) also evidenced the corresponding thermal transitions. On the other hand, it is generally considered that DMA is an important analytical tool for understanding the interphase region of filled polymers. Unfortunately, the effect and magnitude of interfacial interactions are influenced by many factors: the physical properties of the fillers, their chemical properties (the chemistry of the filler, the type of surface treatment, etc.), the chemical and physical properties of the polymer matrix, the quality of filler distribution and dispersion within the matrix, the filler content, etc. [70]. Obviously, the results of DMA allow for the conclusion that increasing the percentage of filler leads to a distinct increase in both E′ and E″ over the full range of investigated temperatures. On the other hand, regarding the differences in relation to the utilization of AIIt and AII-ion, especially when we are comparing PPc–40AIIt and PPc–40AII-ion composites, the ionomeric modification appears to lead to somewhat higher values of E′ and E″ (especially at high temperatures), which can be attributed to the effect of ionomeric crosslinking (presence of Zn clusters) and to stronger interfacial properties. However, in the case of PPc–40AIIt samples, the influence of other factors is not totally excluded, e.g., linked to the presence of 0.4% EBS, which is an additive characterized by a low pre-melting temperature (about 65 °C, see below more comments). Furthermore, these hypotheses agree with the results of HDT testing, providing evidence for a better behavior of PPc–AII-ion with respect to PPc–AIIt composites (see hereinafter).

#### 2.2.6. Heat Deflection Temperature (HDT)

The HDT (B) of the unfilled/nucleated PPc is found to be about 87 °C (Figure 11), whereas the addition of 20–40% AIIt into PPc leads to a slight increase in HDT to about 100 °C, apparently with minor differences between PPc–20AIIt and PPc–40AIIt.

In fact, this can be due to the presence of EBS (0.4% in PPc–40AIIt), which according to DSC analysis shows an initial (secondary) melting peak at about 65 °C, which is ascribed to some excess of stearic acid (used for the reaction with ethylenediamine to produce EBS), whereas the main T_m_ of EBS is determined at 145 °C. On the other hand, the most important enhancement revealed by HDT is obtained for PPc–AII-ion composites, i.e., a substantial increase in HDT up to a temperature of about 112 °C. Clearly, these composites are more recommended to produce engineering components requiring enhanced dimensional stability at high temperatures.

### 2.3. Potential Applications and Further Prospects

As mentioned elsewhere, the applications of PPc and of PPc composites are determined by their specific properties, and they can go from the production of films, sheets, packaging items, transport pallets, containers, pipes, housewares, etc., to components for the automotive industry, mechanical and electrical parts [13,75]. However, the most important difference with respect to PPh is the higher impact strength of PPc (particularly at low temperatures), and this is paving the way for a wide range of applications. Keeping in mind the novelty of the composites concerned by this study, it was also of interest to illustrate their visual aspect after processing by IM as testing specimens (Figure 12a–e), with a mention of the good aesthetic/quality of surfaces and their higher lightness/whiteness. Following the CIELab color measurements, only the values of lightness (L*) are directly indicated in Figure 12 for each composition. By comparing the unfilled PPc (L* = 46.7, milky/opaque aspect, tradeoff on transparency due to the presence of EPR phase), reinforcing with AIIt or AII-ion leads to composites showing higher L* values, which are mainly determined by the percentage of filler, e.g., L* ≅ 75, by filling with 40% AIIt or AII-ion.

To the best of our knowledge, the PPc–AII composites developed in this study are new; therefore, one of our first goals was to focus the attention on the performances of these “tailored” products and on the main findings. Nonetheless, following the application’s requirements, further experimental fine tuning is highly recommended to find and select the optimal processing conditions and compositions. However, more comprehension will be required to explain/analyze the proprieties of these novel composites using additional methods and techniques of characterization (i.e., rheological analysis with rotational rheometers, polarized optical microscopy (POM), X-ray diffraction (XRD), TGA/FTIR/MS coupled techniques to evaluate their thermal degradation by pyrolysis, burning tests, etc.). For instance, the results of primary rheological characterizations (melt flow rate) obtained using melt flow testers and short comments are shown in the Appendix A.

Some more specific characterizations required by the final application could be further necessary, e.g., mechanical testing at low temperatures, evaluation of aging at elevated or low temperatures, under UV conditions, tests of permeability, abrasion, scratch resistance, and so on. Finally, following different experimental strategies for the modulation of interfacial properties, it comes out that the PPc/AII composites can be designed to show improved thermal stability and stiffness (rigidity) together with a good balance between the tensile/flexural strength and impact resistance, which are features enabling them to meet the requirements of various applications.

## 3. Materials and Methods

### 3.1. Materials

The PP copolymer used as a polymeric matrix (PP 402 CB12, producer INEOS Olefins & Polymers Europe; Customer Service Centre: INEOS GmbH, Köln, Germany; Headquarters: Knightsbridge, London, UK) is an anti-static and nucleated high-impact modified PP grade designed for injection-molding applications. Selected properties obtained from the technical sheet of product are as follows: melt flow rate (230 °C, 2.16 kg) = 12 g/10 min; tensile strength at yield = 25 MPa, Izod impact strength = 6 kJ/m^2^ and 10 kJ/m^2^ (at −20 °C and +23 °C, respectively). For simplicity, PP 402 CB12 was abbreviated as “PPc”.

CaSO_4_ β-anhydrite II (AII) delivered as TOROWHITE Ti-ExR04 was kindly supplied by Toro Gips S.L., Zaragoza, Spain. Selected SEM images at low and high magnification are displayed in Figure 13a,b to highlight the shape of AII microparticles [43]. Dynamic light scattering (DLS) was used to analyze the granulometry of AII. The AII microparticles are characterized by a D_v50_ of 3.6 µm and a D_v90_ of 12.9 µm. More specific information about this filler can be also found elsewhere [40,76].

N,N′-Ethylenebis(stearamide) (EBS) delivered as Crodamide EBS by Croda Polymer Additives (Croda Europe Limited, Cowick Hall, Snaith, Goole, East Yorkshire, UK) is a multifunctional processing additive used as a dispersing agent or internal/external lubricant for benefits in plastic applications to facilitate the melt compounding and to improve the dispersion of fillers, to enhance processability, to decrease friction and abrasion at polymer surfaces, etc. Characteristics according to the technical sheet: (main) melting point: 143–149 °C; amine value: 0–3 mg KOH/g; acid value: 0–7 mg KOH/g; Gardner color: 0–4.

Zinc diacrylate (ZA) produced as Dymalink^®^ 9200 by Cray Valley—TotalEnergies (TOTAL Cray Valley, Exton, PA, USA) is an ionomeric monomeric additive recommended for the modification of polyolefins, including PP. This reactive modifier can lead to the increase in melt strength, enhanced mechanical properties and high-temperature performance [51,77,78]. The technical sheet included the following characteristics: off-white powder, 100% active; molecular weight: 207; specific gravity: 1.68; functionality: 2. The chemical structures of the two modifiers (EBS (CAS Number: 110-30-5) and ZA (CAS: 14643-87-9)) are shown in Figure 2.

### 3.2. Production of Mineral Filled PPc Composites

#### 3.2.1. Melt Compounding with Internal Mixers

For a first insight, the filler (AII) was used without any modification. Then, to tune up the properties of composites, the use of special modifiers was considered. The PPc–AII composites have been produced and processed by compression molding (CM) using similar conditions as detailed in our previous work [43]. The plates produced by CM were used to obtain specimens for mechanical characterizations.

#### 3.2.2. Melt Compounding/REx Using TSEs

As illustrated in Figure 14, ZA or EBS were previously premixed with the filler (AII) in a Zeppelin Reimelt Henschel FML4 mixer (Zeppelin Systems GmbH—Henschel Mixing Technology, Kassel, Germany). The mixture (AII/ZA or AII/EBS) was stirred at 1500 rpm for 15 min. In simple terms, the filler was pre-mixed with ZA powder (AII/ZA weight ratio of 20/2 or 40/2, respectively) to produce PPc filled with 20% and 40% AII, containing 2% ZA. In the case of EBS used as a surface modifier/dispersant and processing additive (1 wt.% relative to filler in this study), following its premixing with AII, a supplementary thermal treatment in an oven at 160 °C for 2 h was performed, to allow the melting and better EBS fixation on the surface of particles [46,79].

The mineral-filled PPc composites were produced by melt compounding using a Leistritz twin-screw extruder (TSE) as equipment (ZSE 18 HP-40, Leistritz Extrusionstechnik GmbH, Nürnberg, Germany). Before compounding, the PPc and the fillers were dried overnight in drying furnaces with hot air circulation at 70 °C and 100 °C, respectively. The dosing of PPc (granules) and blended powders of AII/ZA or AII/EBS in TSEs was performed using two different gravimetric feeders. The temperatures of melt compounding/REx were as follows: Z1 = 180 °C; Z2 = 195 °C; Z3 = 215 °C; Z4 and Z5 = 220 °C; Z6 = 210 °C and Z7 = 185 °C; the temperature of the die of extrusion = 180 °C. Other parameters of interest: speed of the screws =170 rpm; throughput = 3 kg/h.

For processing by injection molding (IM), the samples as granules have been dried overnight at a temperature of 70 °C. A Babyplast 6/10 P IM machine (Rambaldi + CO.I.T. s.r.l., Molteno, Italy) was utilized to produce the specimens for the tensile, flexural, impact, and HDT characterizations. Typical temperatures of IM that have been used: Z1 = 205 °C; Z2 = 210 °C; Z3 (die) = 205 °C; temperature of the mold =35–40 °C.

### 3.3. Methods of Characterization

(a) Differential Scanning Calorimetry (DSC): A DSC Q200 from TA Instruments-Waters LLC (New Castle, DE, USA) was used to perform measurements under nitrogen flow. The standard DSC protocol was as follows: a heating scan (10 °C/min) from 0 to +220 °C, an isotherm at this temperature for 2 min, a cooling scan (10 °C/min) to −50 °C, and finally a second heating scan (by 10 °C/min) from −50 to +220 °C. The initial thermal history of the polymer samples was eliminated during the initial scan. TA Instruments Universal Analysis 2000 software-Version 3.9A (TA Instruments—Waters LLC, New Castle, DE, USA) was used to quantify the events of interest, such as those related to the crystallization of PPc during DSC cooling scans, i.e., the peaks of the crystallization temperature (T_c_) and the enthalpies of crystallization (ΔH_c_). It is worth mentioning that the results were normalized to consider only the amounts of PPc in the samples. The melting peak temperature (T_m_), melting enthalpy (ΔH_m_), and final DC (χ) were the thermal parameters that were examined in the second DSC heating scan. The following general equation was used to calculate the DC (degree of crystallinity):
(1)$$\chi =\frac{\Delta H_{m}}{\Delta H_{m}^{0}\times W_{\mathrm{PP}}}\times 100(\%)$$

ΔH_m_ is the melting enthalpy; W_PP_ is the polymer weight fraction in composites; and $\Delta H_{m}^{0}$ is the melting enthalpy of 100% crystalline PP reported to be 207 J/g [80].

(b) Thermogravimetric analyses (TGA): Using a TGA Q50 (TA Instruments, New Castle, DE, USA), the samples were heated under air from room temperature (RT) up to a maximum of 800 °C (platinum pans, 20 °C/min heating ramp, 60 cm^3^/min air flow). The same software was used, i.e., TA Instruments Universal Analysis 2000.

(c) Mechanical testing: Tensile tests were carried out using a Lloyd LR 10K bench machine (Lloyd Instruments Ltd., Bognor Regis, UK) in accordance with the ASTM D638-02a norm on specimens of type V at a speed of 10 mm/min (specimens of 3.0–3.2 mm thickness). NEXYGEN™ MT Materials Test and Data Analysis Software, Batch Version 4.5.1 (Lloyd Instruments Ltd., Bognor Regis, UK), was used to configure the tests, analyze them, and report the results. The nominal strain was determined as the change in grip separation relative to the original grip separation, which was expressed as a percent. The values of Young’s modulus were obtained directly via software and validated for accuracy considering two methods: the secant modulus at low deformation (between 0.05% and 0.25% strain) and the initial tangential modulus (values reported as Young’s modulus).

A three-point bending test and the same NEXYGEN program (Lloyd Instruments Ltd.) were used to configure the test and determine the flexural properties of polymer samples. The Lloyd LR 10K tensile bench (Lloyd Instruments Ltd.) was modified with bending grips (span = 60 mm) in accordance with ISO 178, while the measurements were performed on rectangular specimens (80 × 10 × 4 mm^3^) at a testing speed of 10 mm/min.

Impact resistance: In accordance with the ASTM D256 standard (method A; 3.46 m/s impact speed; 0.668 kg hammer), a Ray-Ran 2500 pendulum impact tester and a Ray-Ran 1900 notching apparatus (Ray-Ran Test Equipment Ltd., Warwickshire, UK) were used to characterize the Izod impact resistance of notched samples (rectangular specimens, 63 × 12 × 3.2 mm^3^). The results reported as energy lost per unit cross-sectional area at the notch (in kJ/m^2^) are calculated by the software of the machine (Ray-Ran 2500).

For all mechanical tests (tensile, flexion, impact), the data were averaged across at least five measurements. The specimens had been conditioned before testing for at least 48 h at 20 ± 2 °C and 50 ± 3% relative humidity.

(d) Scanning Electron Microscopy (SEM): An FE-SEM SU-8020 Hitachi instrument (Hitachi, Tokyo, Japan) with triple detectors was used for the investigations. The analyses were carried out at various accelerated voltages and magnitudes, on cryo-fractured PPc samples at a liquid nitrogen temperature, and on specimens fractured by tensile and impact testing. Secondary electron imaging (SE) and backscattered electron imaging (BSE) detectors were used for the SEM analyses. The reported microphotographs are typical morphologies that have been seen in at least three different places.

(e) Fourier Transform Infrared (FTIR) spectra were recorded on selected samples using a Bruker Tensor 27 FT-IR Spectrometer (Bruker Optics GmbH & Co. KG, Ettlingen, Germany), in the range 800–4000 cm^−1^, with a resolution of 2 cm^−1^. Data were averaged over a total of 40 scans.

(f) The HDT/Vicat 3-300 Allround A1 (ZwickRoell Gmbh & Co., Ulm, Germany) apparatus was used to measure the heat deflection temperature (HDT) in accordance with the ISO 75 norm on three specimens (80 mm × 10 mm × 4 mm) as obtained by IM (“dry” state). Conditions of analysis: load of 0.45 MPa, heating rate of 120 °C/h.

(g) Dynamic Mechanical Analysis (DMA) was carried out utilizing a DMA Q800 device (TA Instruments, New Castle, DE, USA) in dual cantilever bending mode on rectangular specimens obtained by IM (63 × 12 × 3 mm^3^). At a heating rate of 3 °C/min and a frequency of 1 Hz, in the temperature range from −75 to +150 °C, the dynamic storage and loss moduli evolution (E′ and E″, respectively) were analyzed on three specimens using TA Universal Analysis 2000 software (TA Instruments—Waters LLC).

(h) Color measurements (L*, a*, and b*) were made using a SpectroDens Premium (TECHKON GmbH, Königstein, Germany) in the CIELab mode (illuminate D65, 10°). The analyses were accomplished on three specimens by performing at least five measurements on each specimen. Only L* values have been mentioned in the paper.

## 4. Conclusions

To obtain novel mineral-filled composites characterized by improved toughness and increased stiffness, an impact PP copolymer (PPc) was filled with up to 40% CaSO_4_ β-anhydrite II (AII), a filler made from natural gypsum. In the prior tests, the PPc–AII composites were produced with internal mixers and using AII (without any modifier). In correlation with the percentage of filler, some mechanical properties increased (e.g., stiffness) or decreased (important failing of tensile and impact strengths at high filling).

Two key methods have been considered to tune up the properties of composites and modulate their interfacial characteristics: (a) the ionomeric modification of PPc–AII composites by REx using ZA, which is an additive that is able to promote the formation of Zn ionic crosslinks and increase PPc polarity; and (b) the melt compounding of PPc with AIIt previously surface modified with EBS, which is a multifunctional additive. The PPc composites produced with TSEs have been characterized from the viewpoint of morphology, thermal, and mechanical properties, highlighting the differences linked to the amounts of filler (AIIt and AII-ion) and to the key role of interfacial properties (PPc–filler). Compared to the unfilled PPc, the composites showed enhanced characteristics (better thermal stability, increased stiffness evidenced by both tensile and flexural tests, by DMA, etc.).

Two categories of products are obtained, showing distinct characteristic features:PPc–AIIt composites, characterized by good/moderate tensile strength, having as key advantages the high/good ductility (strain at break from 50% to 260%) and a remarkable impact toughness (4.3–5.3 kJ/m^2^) because of the EPR phase and easy debonding of well-dispersed AIIt microparticles.PPc–AII-ion composites, characterized by advanced thermal properties (best values for T_5%_ (TGA), higher HDT), better stiffness and tensile strength (i.e., 28 MPa, at 20–40% AII-ion loading), but low ductility, due to the ionomeric modification and stronger interfacial interactions PPc–filler. Their impact resistance remained at a very interesting level (5 kJ/m^2^), especially at moderate filling (20 wt.%), while a diminishing was noted at higher percentages of AII-ion (3.3 kJ/m^2^).

Characterized by high whiteness and good processing ability, these PPc/AII composites can be ‘tailored’ with specific features to follow the requirements of the applications.

## Data Availability

Not applicable.

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
