# Peer review of "Balancing the Strength–Impact Relationship and Other Key Properties in Polypropylene Copolymer–Natural CaSO4 (Anhydrite)-Filled Composites"

_ijms, 2023, doi:10.3390/ijms241612659_

Round 1

Reviewer 1 Report

The paper describes modifications of PP copolymer by fillers.

The experimental data are well presented and solid. The conclusions are properly supported by the data.

In my opinion the paper can be published as such.

Author Response

We would like to thank the Reviewer for the time used for reading and reviewing our manuscript,
for the appreciation of the results and the amenable comments. The topic of research and the novel results
are of interest for academic and industrial researchers, the study being a direct answer to current industrial
requests regarding the production of new filled PP copolymer-based composites using natural gypsum
derivatives (i.e., AII) as filler. By considering the overall features of these novel composites, it is expected that
they will show interest for various applications (going from packaging to automotive and engineering
components), as new “tailored” materials characterized by stiffness, good balance between the impact
resistance and tensile strength, heat resistance, dimensional stability, cost savings, and other benefits

Reviewer 2 Report

This paper is in large parts a replica of an earlier paper in "Polymers" - see https://doi.org/10.3390/polym15040799. In order to become acceptable, the paper would need to be cleaned up and the previously presented parts removed or used as reference only.

Author Response

We would like to thank the Reviewer for the time spent reading and reviewing our manuscript, quotation, and all suggestions. We also acknowledge for the general appreciation of the scientific results disclosed in this paper, even though our initial manuscript has not received maximum quotes (compared to
most other Reviewers’ Reports), which we interpret as a high level of expectation on the part of the Reviewer. We also express our gratitude for the helpful suggestions made to the authors for improving the work. Please consider below our answers to all comments and suggestions.

R2: Comments and Suggestions for Authors
This paper is in large parts a replica of an earlier paper in "Polymers" -
see https://doi.org/10.3390/polym15040799.
In order to become acceptable, the paper would need to be cleaned up and the previously presented parts
removed or used as reference only.

Authors: We thank the Reviewer for the comments and suggestions. In fact, this is an opportunity for us to improve the manuscript, and to add some more comments about the originality of the novel study/results, compared to the prior art, and mainly related to our very recent paper published in "Polymers"/MDPI / Murariu, M.; Dubois, P. et al., Engineering polypropylene-calcium sulfate (anhydrite II) composites: The key role of zinc ionomers via reactive extrusion. Polymers 2023, 15, 799 (NB: mentioned as one key reference).

First, we would like to point out that the experimental results developed in this study vs. the mentioned paper, are issued in the frame of the same industrial R&D project, having as main goal production of new polymer composites using gypsum derivatives. Inherently, like in the previous paper, one key raw
material is remaining AII as filler, the field of research is the same (i.e., mineral-filled composites based on different polymer matrices), whereas some methods/techniques of production, equipment, and characterization methods, can show (more or less) some similarities.
Is it the research novel enough? Following the feedback to our manuscript, it comes out that maybe we have not arrived well to evidence the originality and added value of our new contribution. We would like to point out the key differences by compared to the Prior Art (including our previous paper):
a)   As it is mentioned in the introductory part, “There are only few references regarding the use of calcium sulfate (CS) to produce PP composites, mostly concerning the reinforcing of PP with synthetic CS whiskers, composites produced in the frame of academic research”, including our recent paper.

The mentioned paper is regarding the production of PP homopolymer (PPh) based composites filled with natural
AII and it valorizes results from the same R&D project, as for this study.
b)   Please take into account that the new study is dealing with the use of PP copolymers (PPc) as a polymer matrix, rather than with PPh, and in our opinion, this is one key difference. As mentioned in the manuscript, “the properties of PP filled composites are largely determined by the nature of PP matrix, i.e., PP homopolymer or PP copolymers” “Owing to the higher impact resistance of PPc, the melt-blending of PPc with rigid fillers appears to be a choice of great interest, by considering the large availability of the raw materials.” Moreover, because the paper is addressed to a large category of Readers (e.g., from academic and applied research), considerable attention was given to introduce and characterize the PPc matrix (high- impact PP copolymer) used in the frame of the experimental program.
c)  The main goal of this new contribution is to show/discuss the experimental pathways followed to
produce, “tailor”’ and characterize novel PPc/AII composites for use in applications requiring materials
assessing a good balance between mechanical strength and impact resistance (NB: higher than using PPh as polymer matrix), improved stiffness, enhanced thermal properties, and other specific benefits.
Inherently, some experimental pathways, the equipment, and the main procedures of characterization remain similar as in the previous study.
d)   Please note that in the prior experimental tests, a dramatic reduction of properties was evidenced (e.g., for the tensile strength and impact resistance), especially at high AII loading. Consequently, two strategies have been considered. As it is mentioned in the manuscript, “For better filler dispersion and compatibility, or stronger interfacial adhesion, custom formulations have been performed, respectively, following the physical/surface modification of AII microparticles with ethylenebis(stearamide) (EBS)
[references], or using reactive metallic ionomeric additives (i.e., ZA) [references], both methods reported in
prior art to improve the properties of polymeric composites.”
e)   Indeed, the reactive modification (REx) of PPc-AII compositions/composites with metallic ionomeric additives (i.e., zinc diacrylate - ZA) has been once more considered to obtain significant enhancements of properties (NB: like in our previous paper). We have preferred to use ZA as modifier/compatibilizer rather than the traditional maleic anhydride–grafted polypropylene (MAgPP) which is considered in many studies.
The confirmation of the key role of ZA as excellent reactive modifier to produce these specific new composites is considered an advancement to the current knowledge. On the other hand, the specific treatment of AII with EBS (NB: not concerned in the previous study) opened the way to a second category of
PPc/AII composites, assessing better ductility and impact resistance. To the best of our knowledge, byconsidering the State of the Art, these results in relation to the production of PPc/AII composites are original/new.
f)    The properties of these ‘tailored’ PPc-AII composites cannot be predicted, their investigation was required: “The effects of filling PPc with up to 40 wt.% AII in these custom composites were deeply evaluated in terms of morphology, mechanical and thermal properties, to evidence their key-properties for potential/further applications.” The use of FTIR analysis for the evidence of modifications of PPc-AII compositions by reactive melt-blending and by reactive extrusion (REx), is an additional contribution to the actual knowledge, not found in our early study.
g) The specificity of these novel polymer composites: the study is proving that the properties of PPc-AII composites can be particularly designed using specific modifiers/compatibilizers for various end-use applications. Hence, we strongly believe that the manuscript depicts original results and will deserve
publication in the Special Issue " Recent Advances in Polymer and Polymer Composites" of IJMS.

Thanking the Reviewer for the comments and suggestions, we have improved the manuscript by highlighting the originality of the novel contribution/products, by removing some parts and by updating the manuscript
as it was suggested.
Since the reordering and restructuring of the manuscript was substantial, we have written bullet points of
our major changes to the manuscript, rather than including a list of ‘track changes’ text:
Abstract: The text was improved, made shorter and clearer, the key properties of interest are
quantified by values.
Introduction: Up-date and more brief presentation. A large part of text was removed, the novelty of
results/research is better evidenced.
Results and Discussion: The manuscript was restructured and updated. We tried our best not to
repeat previous information, the early paper is given as reference. Some parts of text are removed,
some additional explanations are added, including the indication of forthcoming studies to follow
more specific characterizations of PPc/AII composites is mentioned.
Materials and methods: These sections are restructured and changed. In fact, in the initial manuscript, our first intention was to give the Readers the most important experimental information, with the risk of some repetitions (i.e., for methods of analysis), which can increase the “similarity
index”. Following the comment/recommendation of the Reviewer, large parts of text are removed (e.g, Melt compounding with internal mixers.)
The procedure is the same as in the previous study, this part is removed and is given only the
reference for more details.
Conclusions: This section was revised, the text was made shorter, clearer, highlighting the
advantages and disadvantages of two categories of composites developed in this study.

Bellow, we would like to give only a few examples of changes in the revised manuscript:

 Introduction

  • “Related to this study, which valorizes and develops the results of an applied R&D project, it is worth mentioning that nowadays the producers of natural gypsum are looking for new markets and new
    applications...
  • “The reactive modification of PPc-AII compositions with metallic ionomeric additives (i.e., zinc diacrylate - ZA) has been once more considered to obtain more performant polymer composites [43].”
  • "Finally, these results assess for the key role of the polymer matrix and of interfacial properties, highlighting the benefits of special modifiers like ZA and EBS, which are paving the way for two categories of products
    with distinct characteristic features in the specific case of PPc/AII composites.

Results and Discussion

  • Please consider as a starting point: “The tensile and impact strengths dramatically decreased, especially at high filling (40 wt.%).”
  • Supplementary text added in the revised document:
    The decrease of mechanical properties (i.e., tensile strength, impact resistance, etc.) at high loadings is traditionally attributed to the formation of aggregates due to particle/particle interactions, creation of stress-concentration points, to low matrix/filler interactions, ineffective wetting of the filler by the polymer, insufficient adhesion and homogenization, and so on [references].”
  • Linked to thermal properties:
    “Traditionally, the fillers improve the thermal stability of polymer composites (references). The addition of AII into different polymer matrices (e.g., PLA, PP, others) was reported to lead to a delay in their thermal degradation (a significant stabilization effect), which can be quite well distinguished by the increase of the maximum decomposition temperatures (Td) /references.”

Further prospects:
“To the best of our knowledge, the PPc-AII composites developed in this study are new, therefore, one of our first goals was to focus the attention on the performances of these “tailored” products and on the main findings. However, more comprehension will be required to explain/analyze their proprieties using additional
methods and techniques of characterization (i.e., rheological analysis with rotational rheometers, polarized optical microscopy (POM), X-ray diffraction (XRD), TGA/FTIR/MS coupled techniques to evaluate their thermal degradation by pyrolysis, burning tests, etc.). For instance, the results of primary rheological
characterizations (Melt Flow Rate) using melt flow testers and short comments are shown in Supplementary
Material S2: Rheological investigations (MFR) on samples produced with twin-screw extruders (TSE).” ...

3. Materials and Methods: This section has been restructured and significantly changed. Large parts of text
are removed.

Conclusions:
“Two categories of products are obtained, showing distinct characteristic features: (1) PPc-AIIt composites, characterized by good/moderate tensile strength, having as key advantage the high/good ductility (strain at break from 50% to 260%) and remarkable impact toughness (4.3 - 5.3 kJ/m2), because of the EPR phase and easy debonding of well dispersed AIIt microparticles; and (2) PPc-AII-ion composites, characterized by advanced thermal properties (evidenced by TGA, higher Tc and HDT) and stiffness, better tensile strength (i.e., 28 MPa at 20 - 40% AII-ion), but low ductility, due to the ionomeric modification and stronger interfacial interactions. Their impact resistance remained at a very interesting level (5 kJ/m2), especially at moderate filling (20 wt.%), while a diminishing was noted at higher percentages of AII-ion (3.3 kJ/m2).”
...
NB: For shortness, we have considered that it is not necessary to mention here all changes and removed paragraphs (a list of all changes can be obtained if it is necessary).

Authors: Finally, we would like to thank the Reviewer once again for the time spent reading & reviewing our manuscript, and for making suggestions. Following the viewpoints and amenable comments of the Reviewer,
we had the opportunity to point out in our answers and in the revised manuscript the originality of our study. We hope that the Reviewer will appreciate our answers and that the changes made will satisfy most of the
requirements for publication. The paper has been carefully revised, updated, and restructured according to all comments, and all changes have been evidenced using the "Track Changes" function/or yellow ink, directly within the revised contribution.

Reviewer 3 Report

Article "Balancing the strength-impact relationship and other key properties in polypropylene copolymer - natural CaSO4 (anhydrite) filled composites" is devoted to the study of mineral filled polymer composites. This article covers a large amount of research and is very interesting. However, there are a number of minor comments:

1) it would be very interesting to investigate the rheological characteristics of composite melts (e.g., viscosity, storage and loss moduli), as this would directly determine the conditions of their use and operation

2) it would be useful to perform an edx analysis, from which one could judge the uniform distribution of the filler in the polymer composite

3) perhaps X-ray analysis could also further characterize the resulting composites

Reviewer 4 Report

As the International Journal of Molecular Sciences the topic of the work at hand would appear to be an appropriate one, in particular paying attention to the research.

The abstract is a little bit confuse and missis some information like more results and conclusions.

keywords should be reduced

The Introduction section quite briefly refers to the content of the article, of course the authors pay attention to the key theses from the area of literature analysis, but this section should be reduce by a general thematic introduction. 

It would be reasonable for the reader to introduce analysis of the properties of such materials.

most important notes: 

-the research methodology should be described in detail, including the preparation of materials,

- figures should be corrected in accordance with the guidelines of the journal, 

- the text needs editorial correction in accordance with the requirements of the journal and the arrangement of photos in the figures needs to be improved; font, references, etc.

- how many samples were tested?, please refer to the standards for testing properties,

- measurement parameters are not described

- all the test methods used (e.g. machines, devices) are not described in detail (e.g tensile strength and strain at break;  Young’s modulus, Izod impact resistance.

- part of the methodology includes test results (e.g. lightness (L*), are they catalog data?

- the text should be systematized, the methodology separately, the research results separately

- it is necessary to compare the results of FTIR tests with TG, D-TG, describe exactly what gases are generated during mass loss, m/z could be presented

- the test results should be analyzed and the reasons for the changes in properties obtained should be indicated,

The conclusions do not refer to the work, but to the description of what the work presents. It is recommended to conduct a deeper discussion and refer to the results in the conclusions, also critically presenting the advantages and disadvantages of the method - which does not seem to be difficult when reading the paper.

minor editing of English language required

Reviewer 5 Report

The research is focused on composites based on polypropylene copolymer (PPc) containing up to 40% by weight of an additive from natural gypsum. Among other things, the modification of the matrix with zinc dyacrilate ionomer by reactive extrusion and the use of a twin-screw extruder were also considered to improve the dispersion of the additive and the additive-matrix interaction.

The samples prepared by compression molding and injection molding were systematically characterized with different and appropriate analytical techniques in terms of thermal, mechanical (both quasi-static and dynamic), physical properties as well as to highlight morphological aspects.

The manuscript is correctly articulated and the results are clearly reported and adequately discussed also with reference to similar experimental evidence already available in the literature.

Overall, the work allows an advancement of current knowledge by enhancing the opportunity to develop new PPc-based composites, tailor made to satisfy specific applications in various industrial sectors.

The manuscript is considered acceptable for publication as received.

Author Response

The research is focused on composites based on polypropylene copolymer (PPc) containing up to 40% by weight of an additive from natural gypsum. Among other things, the modification of the matrix with zinc dyacrilate ionomer by reactive extrusion and the use of a twin-screw extruder were also considered to improve the dispersion of the additive and the additive-matrix interaction.

The samples prepared by compression molding and injection molding were systematically characterized with different and appropriate analytical techniques in terms of thermal, mechanical (both quasi-static and dynamic), physical properties as well as to highlight morphological aspects.

The manuscript is correctly articulated and the results are clearly reported and adequately discussed also with reference to similar experimental evidence already available in the literature.

Overall, the work allows an advancement of current knowledge by enhancing the opportunity to develop new PPc-based composites, tailor made to satisfy specific applications in various industrial sectors.
The manuscript is considered acceptable for publication as received.

Authors:
We would like to thank the Reviewer for the time used for reading and reviewing our manuscript, for the appreciation of the results and amenable comments. The topic of research and the novel results are of interest for academic and industrial researchers, the study being a direct answer to current industrial requests regarding the production of new filled PP copolymer-based composites using natural gypsum derivatives (i.e., AII) as filler. By considering the overall features of these novel ‘tailored’ composites (stiffness, good balance between the impact resistance and tensile strength, heat resistance, cost savings, other benefits), it is expected that they will show potential interest as new materials that can be designed for various applications, going from packaging to automotive and engineering components.
We thank the Reviewer for the conclusion that “the work allows an advancement of current knowledge”. Hence, we strongly believe that the manuscript depicts original results and will deserve publication in the
Special Issue " Recent Advances in Polymer and Polymer Composites" of IJMS.

Round 2

Reviewer 2 Report

The paper has been improved significantly and is now fine for publishing.